# Karyotypes and Physical Mapping of Ribosomal DNA with Oligo-Probes in *Eranthis* sect. *Eranthis* (Ranunculaceae)

**DOI:** 10.3390/plants13010047

**Published:** 2023-12-22

**Authors:** Elizaveta Yu. Mitrenina, Svetlana S. Alekseeva, Ekaterina D. Badaeva, Lorenzo Peruzzi, Gleb N. Artemov, Denis A. Krivenko, Lorenzo Pinzani, Zeki Aytaç, Ömer Çeçen, Shukherdorj Baasanmunkh, Hyeok Jae Choi, Attila Mesterházy, Alexander N. Tashev, Svetlana Bancheva, Lian Lian, Kunli Xiang, Wei Wang, Andrey S. Erst

**Affiliations:** 1Department of Genetics and Cell Biology, Biological Institute, National Research Tomsk State University, 634050 Tomsk, Russia; emitrenina@gmail.com (E.Y.M.); sveta.alexx@mail.ru (S.S.A.); g-artemov@mail.ru (G.N.A.); 2Central Siberian Botanical Garden, Siberian Branch of the Russian Academy of Sciences, 630090 Novosibirsk, Russia; krivenko.irk@gmail.com; 3Vavilov Institute of General Genetics, Russian Academy of Sciences, 119333 Moscow, Russia; katerinabadaeva@gmail.com; 4PLANTSEED Lab, Department of Biology, University of Pisa, 56126 Pisa, Italy; lorenzo.peruzzi@unipi.it (L.P.); lorenzo.pinzani@uniroma3.it (L.P.); 5Siberian Institute of Plant Physiology and Biochemistry, Siberian Branch of the Russian Academy of Sciences, 664033 Irkutsk, Russia; 6Biology Department, Faculty of Science, Gazi University, Ankara 06500, Turkey; zaytac@gazi.edu.tr; 7Department of Plant and Animal Production, Technical Sciences Vocational School, Karamanoğlu Mehmetbey University, Karaman 70100, Turkey; omercecen@kmu.edu.tr; 8Department of Biology and Chemistry, Changwon National University, Changwon 51140, Republic of Korea; baasanmunkh.sh@gmail.com (S.B.); hjchoi1975@changwon.ac.kr (H.J.C.); 9Independent Researcher, H-9500 Celldömölk, Hungary; amesterhazy@gmail.com; 10Department of Dendrology, University of Forestry, 1756 Sofia, Bulgaria; altashev@mail.ru; 11Botanical Garden, Bulgarian Academy of Sciences, 1616 Sofia, Bulgaria; sbancheva@yahoo.com; 12Institute of Biodiversity and Ecosystem Research, Bulgarian Academy of Sciences, G. Bonchev, Bl.23, 1113 Sofia, Bulgaria; 13State Key Laboratory of Systematic and Evolutionary Botany, Institute of Botany, Chinese Academy of Sciences, Beijing 100093, China; lianlian@ibcas.ac.cn (L.L.); kunlixiang@ibcas.ac.cn (K.X.); wangwei1127@ibcas.ac.cn (W.W.); 14College of Life Sciences, University of Chinese Academy of Sciences, Beijing 100049, China

**Keywords:** *Eranthis*, chromosomes, fluorescence in situ hybridization (FISH), karyotype, oligonucleotide probes, Ranunculaceae, 45S and 5S rDNA clusters

## Abstract

A comparative karyotype analysis of four species of yellow-flowered *Eranthis* sect. *Eranthis*, i.e., *E. bulgarica*, *E. cilicica*, *E. hyemalis*, and *E. longistipitata* from different areas, has been carried out for the first time. All the studied specimens had somatic chromosome number 2*n* = 16 with basic chromosome number *x* = 8. Karyotypes of the investigated plants included five pairs of metacentric chromosomes and three pairs of submetacentric/subtelocentric chromosomes. The chromosome sets of the investigated species differ mainly in the ratio of submetacentric/subtelocentric chromosomes, their relative lengths, and arm ratios. A new oligonucleotide probe was developed and tested to detect 45S rDNA clusters. Using this probe and an oligonucleotide probe to 5S rDNA, 45S and 5S rDNA clusters were localized for the first time on chromosomes of *E. cilicica*, *E. hyemalis*, and *E. longistipitata*. Major 45S rDNA clusters were identified on satellite chromosomes in all the species; in *E. cilicica*, minor clusters were also identified in the terminal regions of one metacentric chromosome pair. The number and distribution of 5S rDNA clusters is more specific. In *E. cilicica*, two major clusters were identified in the pericentromeric region of a pair of metacentric chromosomes. Two major clusters in the pericentromeric region of a pair of submetacentric chromosomes and two major clusters in the interstitial region of a pair of metacentric chromosomes were observed in *E. longistipitata*. *E. hyemalis* has many clusters of different sizes, localized mainly in the pericentromeric regions. Summarizing new data on the karyotype structure of *E.* sect. *Eranthis* and previously obtained data on *E.* sect. *Shibateranthis* allowed conclusions to be formed about the clear interspecific karyological differences of the genus *Eranthis*.

## 1. Introduction

The genus *Eranthis* Salisb. (winter aconite) belongs to Ranunculaceae Juss., tribe Cimicifugeae Torr. & A. Grey [1]. This genus consists of 10 to 14 early flowering herbaceous perennial species that are distributed across Southern Europe and Western, Central, and temperate Asia [2,3]. Most species of the genus have a limited distribution [4,5]. According to morphological data, the genus has been divided into two sections: *E.* sect. *Eranthis* and *E*. sect. *Shibateranthis* (Nakai) Tamura [6]. The type section *Eranthis* is characterized by plants with tubers, yellow sepals, and emarginate or slightly bilobate upper petal margins without pseudonectaries [7]. *Eranthis* sect. *Eranthis* in Europe includes *E. bulgarica* (Stef.) Stef. and *E. hyemalis* (L.) Salisb., whereas in Southwest and West Asia, it includes *E. cilicica* Schott & Kotschy, *E. iranica* Rukšāns & Zetterl., *E. kurdica* Rukšāns, and *E. longistipitata* Regel [8,9,10]. *Eranthis hyemalis* is often associated with estate woodlands. However, though present in many European countries, it is probably only truly native to the south, France, and Italy. In the wild, *E. cilicica* and *E. hyemalis* can be readily separated [11]. However, Stace’s comments would certainly be true for some plants in cultivation and some naturalized populations in Britain and Ireland [12]. Plants with intermediate characters can be found, and these can perhaps be referred to the hybrid *Eranthis* × *tubergenii* Bowles. *Eranthis longistipitata* is a plant of mountain woodlands and part-shaded rocky habitats, occasionally venturing onto grassy slopes and screes, found in Afghanistan, Kazakhstan, Tajikistan, Turkmenistan, and Uzbekistan. Its distribution does not overlap with any other species of *Eranthis* [11]. *Eranthis longistipitata* and *E. cilicica* grow in places that are reasonably humid during the growing season. After the plants die down, their growing places become very dry, and that is exactly what this magnificent plant appreciates in the garden [13]. The more orange-colored plants described as *E. iranica* are common in Iran and at the border of Turkmenistan [9]. *Eranthis kurdica* was described recently [10]. This species was initially classified as *E. cilicica* or *E. hyemalis*, yet it differs in the presence of daughter tubers on a mother tuber, similar to *E. iranica* and *E. longistipitata*. This species is common in eastern Turkey, Iran, and Iraq [14].

*Eranthis* sect. *Shibateranthis* has long-lived tubers, white sepals, and bilobate or forked petal margins with pseudonectaries [7,15]. Species of this section have a natural geographic range in temperate North and East Asia, i.e., *Eranthis albiflora* Franch., *E. byunsanensis* B.Y. Sun, *E. lobulata* W.T. Wang, *E. pinnatifida* Maxim., *E. pungdoensis* B.U. Oh, *E. sibirica* DC., *E. stellata* Maxim., and *E. tanhoensis* Erst [2,4,5].

Chromosomal analysis is extensively used to study the systematics and evolution of plants from different taxonomic groups [5,16,17,18]. In Ranunculaceae, the karyotype structure has been studied for representatives of numerous genera, particularly those with large chromosomes, such as *Adonis* L. [19], *Anemone* L. [20], *Caltha* L. [21,22], *Cimicifuga* Wernisch. [21,22], *Halerpestes* Greene [23], *Helleborus* L. [22], *Ranunculus* L. [24], *Trollius* L. [21], etc. However, molecular cytogenetic studies, an integral part of chromosomal analysis, were conducted only for some taxa of this family. Similar studies on the localization of conserved ribosomal DNA sequences are known for *Anemone* [19], *Hepatica* Mill. [25], *Pulsatilla* Mill. [26,27], and *Ranunculus* [28]. 

In our previous study, we described the chromosome sets of six species of *Eranthis* sect. *Shibateranthis* and revealed clear karyotypical differences in *E. byunsanensis*, *E. lobulata*, *E. pinnatifida*, *E. sibirica*, *E. stellata*, and *E. tanhoensis* associated with their evolution. The studied species differ in the number and morphology of chromosomes [29]. For the species of *E.* sect. *Eranthis*, only the karyotype of *E. hyemalis* was described [30,31,32]. The purpose of our current study was to perform comparative analysis of karyotype structure and physical mapping of 45S and 5S ribosomal DNA (rDNA) sequences on chromosomes by fluorescence in situ hybridization (FISH) in four species *Eranthis* sect. *Eranthis*, i.e., *E. bulgarica*, *E. cilicica*, *E. hyemalis*, and *E. longistipitata* (Figure 1). Highly conserved rDNA sequences are widely used for comparative molecular cytogenetic analysis of different taxa to study the evolution of chromosome sets [33,34]. 

## 2. Results

### 2.1. Karyotypes of Eranthis sect. Eranthis

Comparative karyological analysis was performed for *Eranthis bulgarica*, *E. cilicica*, *E. hyemalis*, and *E. longistipitata* from 18 populations from different areas (Table 1). The investigated accessions showed predominantly large and medium-sized chromosomes that belong to the *Ranunculus*-type according to Langlet [35]. The basic and somatic chromosome numbers were *x* = 8 and 2*n* = 16, respectively (Figure 2). Their karyotypes included five pairs of large metacentric chromosomes and three pairs of smaller submetacentric and subtelocentric chromosomes in different ratios. Karyotype formulas for accessions from 13 populations are summarized in Table 2. The results of a detailed karyomorphometric analysis and haploid idiograms for accessions from four populations (one for each species) are presented in Table 3 and Figure 3, respectively.

#### 2.1.1. *Eranthis bulgarica*

Metacentric chromosomes in this species formed a homogeneous group that gradually decreased in length and had similar arm ratios (Figure 2A,B and Figure 3; Table 3). The relative lengths of individual metacentric chromosomes in this species varied within a range from 6.20 to 7.75%. The other chromosomes included three submetacentric and three subtelocentric ones. The longest chromosome pair was submetacentric (mean relative length of 5.41%) and the shortest chromosome pair was subtelocentric (mean relative length of 4.45%) with small satellites in the terminal region of the short arm. Typically, one satellite chromosome and less often two satellite chromosomes could be identified on metaphase plates. The third pair of non-metacentric chromosomes in accessions from pop. 1 was of intermediate size; it was heteromorphic and was represented by submetacentric and subtelocentric chromosomes of a similar length. In accessions from pop. 2, this chromosome pair differed not only in the arm ratio, but also in length, i.e., the mean relative length of the submetacentric chromosome was 2.92%, and the mean relative length of the subtelocentric chromosome was 5.19% (Figure 2B). The karyotype formula of *E. bulgarica* was 2*n* = 2*x* = 16 = 10m + 3sm + 1st + 2st^sat^.

#### 2.1.2. *Eranthis cilicica*

We studied the karyotype structure of accessions from two populations growing in Turkey (Figure 2C). The karyotype formulas were similar for both populations—2*n* = 2*x* = 16 = 10m + 4sm + 2st^sat^. A more detailed karyomorphometric analysis was carried out for pop. 3 from Kahramanmaraş Province (Figure 3; Table 3). Five pairs of metacentric chromosomes differed slightly in the arm ratios and lengths (mean relative length of 6.35 to 7.60%). Two other pairs were submetacentric chromosomes and one pair was subtelocentric chromosomes with small satellites in short arms. The length of one pair of submetacentric chromosomes was very close to the length of subtelocentric chromosomes, whereas the second pair of submetacentric chromosomes was significantly shorter (mean relative lengths of the two pairs of submetacentric chromosomes were 5.49 and 3.40%).

#### 2.1.3. *Eranthis hyemalis*

We studied chromosome sets of accessions from six populations growing in Italy, Germany, and Hungary (Figure 2D–F; Table 2). The karyotype formulas were similar for all the studied plants, i.e., 2*n* = 2*x* = 16 = 10m + 2sm + 2st + 2st^sat^. A detailed karyomorphological analysis was carried out for the population from Via di Roncrio, Bologna, Italy (Figure 3; Table 3). Similar to other species, five pairs of metacentric chromosomes were the largest in the set; the arm ratios and chromosome lengths varied (mean relative length of 6.46 to 7.90%). The other pairs of chromosomes had similar lengths (mean relative length of 4.51 to 4.84%), but differed in the arm ratios. A more asymmetrical subtelocentric pair carried small satellites in the terminal regions of the short arms.

#### 2.1.4. *Eranthis longistipitata*

We studied the karyotype structure of accession from three populations growing in Kazakhstan, Tajikistan, and Uzbekistan (Figure 2G–I; Table 2). The karyotype formula for *E. longistipitata* was 2*n* = 2*x* = 16 = 10m + 2sm + 2st + 2st^sat^. A detailed karyomorphological analysis was carried out for the accession from Kazakhstan (Table 3). The arm ratios and lengths varied in five pairs of metacentric chromosomes (mean relative lengths of 6.61 to 8.27%). The shortest heterobrachial chromosome pair was submetacentric, and its average relative length attained 3.52%. Two pairs of subtelocentric chromosomes differed in lengths (mean relative lengths of 4.79 and 4.12%), but their arm ratios were similar (Table 3). A pair of shorter subtelocentric chromosomes had small satellites in the terminal regions of the short arms.

#### 2.1.5. Comparative Karyomorphometric Analysis

The exploratory PCA (cumulative variance explained by the first two axes: 90.34%) highlighted four non-overlapping groups (Figure 4). This was further confirmed by univariate analyses: THL values (ANOVA, F = 71.66, df = 3, *p* << 0.01) differed among all species pairs with *p* << 0.01 (Tukey HSD test), with the exception of the pair *E. bulgarica* vs. *E. longistipitata*, which did not show significant differences (*p* = 0.71). CV_CL_ values (F = 61.13, df = 3, *p* << 0.01) differed among all species pairs with *p* << 0.01, with the exception of the pair *E. cilicica* vs. *E. hyemalis* (*p* = 0.98). M_CA_ values (F = 4.96, df = 3, *p* << 0.01) differed only between *E. bulgarica* and *E. hyemalis* with *p* << 0.01. Finally, CV_CI_ values (F = 27.59, df = 3, *p* << 0.01) differed among all species pairs with *p* << 0.01, with the exception of the pair *E. cilicica* vs. *E. longistipitata* (*p* = 0.99). Consequently, the highest values of THL and CV_CI_ were shown by *E. hyemalis*, the highest values of CV_CL_ by *E. longistipitata* (and the lowest by *E. bulgarica*), while *E. cilicica* showed intermediate values concerning all features (Table 3). The overall results were fully supported by LDA, that correctly attributed object (accessions) to the four species in 100% of cases (jackknifed).

### 2.2. Distribution of rDNA Sites along the Chromosomes of Eranthis cilicica, E. hyemalis, and E. longistipitata

FISH mapping of 45S and 5S rDNA clusters on chromosomes of the *Eranthis* species was performed for the first time. We studied *E. cilicica* (pop. 3), *E. hyemalis* (pop. 10), and *E. longistipitata* (pop. 17) (Figure 5) and revealed that all the species had two major clusters of 45S rDNA (NORs) at the terminal regions of short arms of subtelocentric satellite chromosomes. In *E. cilicica*, 1–2 pairs of minor clusters of 45S rDNA were identified in the terminal regions of metacentric chromosomes.

The studied species showed differences in the number of 5S rDNA clusters. In *E. cilicica*, only two major clusters of 5S rDNA were identified in subcentromeric regions of the shortest metacentric chromosomes (Figure 5A). *Eranthis longistipitata* had four major clusters of 5S rDNA—in the interstitial regions of a pair of metacentric chromosomes and in the pericentromeric regions of a pair of small submetacentric chromosomes (Figure 5C). *Eranthis hyemalis* had up to 16 5S rDNA clusters of various sizes in 14 chromosomes (Figure 5B). They were mainly localized in the pericentromeric regions, but a pair of submetacentric chromosomes without satellites had two clusters each—in the pericentromeric and interstitial regions.

## 3. Discussion

### Karyotype Structure in Eranthis sect. Eranthis

Our current and previous karyological studies on *Eranthis* showed that the genus was predominantly characterized by a basic chromosome number *x* = 8 and diploid cytotypes with 2*n* = 16. The exceptions included records of triploid *E. hyemalis* with 2*n* = 24 [36], and two species from Siberia that belonged to *E.* sect. *Shibateranthis*, namely *E. tanhoensis* with 2*n* = 14 and *E. sibirica* with 2*n* = 42 [5,29]. The karyotype structure of the studied species—*E. bulgarica*, *E. cilicica*, *E. hyemalis*, and *E. longistipitata*—showed common features within this group and shared with some species of *E.* sect. *Shibateranthis*, namely *E. byunsanensis*, *E. lobulata*, and *E. stellata*. The chromosome sets of these species included five pairs of large metacentric chromosomes and three pairs of smaller chromosomes with unequal arms of different morphology. The chromosome set of each of the species included a pair of satellite submetacentric or subtelocentric chromosomes with a secondary constriction on the short arm. This type of localization of the nucleolar organizer (in the subterminal region of the short arm) is characteristic of karyotypes of most plants [37]. We identified a different localization of the secondary constriction for *E. sibirica* and *E. tanhoensis*—in the interstitial region of one of the arms of metacentric chromosomes [29]. It is known that the secondary constrictions represent only the expression of rRNA genes, which were active during the preceding interphase, and other functional sites may not form secondary constrictions, especially if located too close to the terminal region of the chromosomes. The 45S rDNA clusters, which do not form secondary constrictions, can be identified using the FISH technique [33]. Despite similarities in karyotype, each of the studied *Eranthis* species showed its own species-specific features. For example, unlike *E. hyemalis* and *E. bulgarica*, the karyotype of *E. cilicica* and *E. longistipitata* included a pair of very short submetacentric chromosomes (pair 8). In turn, the karyotypes of *E. cilicica* and *E. longistipitata* differed in the morphology of chromosomes with unequal arms. Compared to the three studied species, a distinctive feature of *E. hyemalis* was the presence of the largest chromosomes. Despite the fact that lengths of chromosomes were affected by the method of plant material pretreating and the degree of chromosome condensation [38], it was obvious that *E. hyemalis* and *E. longistipitata*, having similar karyotype formulas, showed apparent differences in the total length of haploid chromosome set.

Confusion surrounds the circumscription of *Eranthis hyemalis* and *E. cilicica*—should they be considered as a single species or two? Checking regional flora and other accounts, these two taxa were treated in different ways by different authors, but close study revealed the prime elements involved were very different, not only in their morphology but also in their geography and preferred habitats [11]. The flowers of *E. cilicica* are more golden and the leaves are more finely pinnate. When they start to grow, the leaves also acquire a more bronze color, which fades during leaf unfolding. The tuber of *E. cilicica* is round and smooth, whereas the tuber of *E. hyemalis* is knobby and uneven. In most cases, it flowers later than *E. hyemalis* [13]. The hybrid between the two species (*Eranthis* × *tubergenii*) bears intermediate characteristics but is generally more vigorous and substantial, possessing a high degree of sterility, which is a further indication that *E. cilicica* and *E. hyemalis* are specifically distinct [11].

However, the population of winter aconite at peak Vraška Čuka (on the border between Bulgaria and Serbia) was subsequently identified as *E. bulgarica* Stef. [39,40], which grows on karst and lithosol, in the xerothermal belt of oak forests [41]. This enigmatic species shows minor morphological differences from populations growing in Italy (the locus classicus for *E. hyemalis*) and requires further studies using an integrative taxonomical approach, including cytogenetic methods. In this study, we clearly showed that *E. bulgarica* and *E. hyemalis* differed in the karyotype structure. *Eranthis bulgarica* showed a heteromorphic chromosome pair, which consisted of submetacentric and subtelocentric chromosomes. We previously observed a similar feature in the karyotype of *E. byunsanensis* [29]. However, we could not assert that such karyotype structures were characteristic of this species throughout its entire range since we examined the karyotype from a single locality. A similar length of the heteromorphic chromosomes in *E. bulgarica* (pop. 1) suggests the presence of a pericentric inversion that changed the morphology of one of the chromosome pairs. However, a difference in their lengths in one of the accessions (pop. 2) suggests a more complex chromosomal rearrangement, presumably translocation. A similar translocation was observed in *Adonis apennina* L. (indicated as *A. sibirica* (*Patrin ex DC.*) *Spach*) from one of the Siberian populations [42] and in *Adonis vernalis* from one of the populations in the Altai Territory (Russia) [43].

We studied the karyotypes of *E. bulgarica*, *E. cilicica*, and *E. longistipitata* for the first time, whereas data on the karyotype of *E. hyemalis* were previously reported by Tak and Wafai [30], Gömürgen [31], and Caparelli, Aquaro, and Peruzzi [32]. According to [30], the chromosome set of this species was 2*n* = 16 = 10m + 6st and the karyotype structure, namely the ratio of metacentric/non-metacentric chromosomes, was close to that reported in our study. According to [31], the karyotype formula for *E. hyemalis* is 2*n* = 16 = 12m + 4sm, which differs from our data. Differences in karyotype formulas were not due to differences in chromosome nomenclature, since the author followed the same classification coined by Levan et al. [44]. 

We examined karyotypes of *E. hyemalis* from six localities, including four localities in Italy, which was the locus classicus for this species [45]. All the investigated plants shared a similar karyotype with clearly identifiable three pairs of submetacentric and subtelocentric chromosomes. The karyotype formula of the *Eranthis* accession from Turkey (Afyon region) studied by Gömürgen [31] differed from the karyotype formula of *E. cilicica* from Turkey included in the current study. Apparently, the material investigated by Gömürgen belonged to taxon other than *E. hyemalis* or *E. cilicica* since new taxa are still being reported [9,10], including those from this region (Turkey) [14]. In another study [32] conducted for *E. hyemalis* from Italy, the karyotype formula was determined as 2*n* = 16 = 8m + 4sm + 2st + 2t^sat^. 

The karyotype structures of *E. bulgarica*, *E. cilicica*, *E. hyemalis*, and *E. longistipitata* were also similar in the intrachromosomal asymmetry calculated by mean centromeric asymmetry (M_CA_) [46]. However, the values of the coefficient of variation in chromosome length (CV_CL_) [47], which indicated the levels of interchromosomal asymmetry, varied in a greater degree within the group (Table 3). The differences between *E. bulgarica* and *E. longistipitata* were noticeable; the latter species showed a more asymmetric karyotype.

The results obtained in the current and our previous study [29] showed that *E.* sect. *Eranthis* was karyologically more homogeneous than *E.* sect. *Shibateranthis*. The latter clearly showed two subgroups of species according to the basic number of chromosomes, and structure and asymmetry of karyotypes. The karyotype structure and asymmetry are the features that change during evolution. The metacentric and submetacentric chromosomes of the symmetrical karyotype are approximately equal in size. Changes towards an asymmetric karyotype can arise by shifts in centromere position (intrachromosomal asymmetry) and/or by the addition or deletion of chromatin from some but not all chromosomes, leading to differences in size between the largest and smallest chromosomes (interchromosomal asymmetry) [17]. Most genera of herbaceous angiosperms displayed interspecific differences in chromosome size and symmetry, if not number [48]. The karyotypes in Ranunculaceae are generally considered to have evolved towards increasing asymmetry, as was reported by Lewitsky [49] based on his cytological study of the tribe *Helleboreae* s.l. He believed that, in this and other groups, chromosomes became more asymmetrical as evolution progressed. Evolution can demonstrate the opposite process, namely, an increase in the karyotype symmetry due to Robertsonian chromosome fusions and other rearrangements [50]. Thus, classical karyotyping methods are not sufficient to establish the direction of changes; it requires an integrative approach to the study of karyotype evolution, including molecular phylogenetic and molecular cytogenetic analyses [24,33,34,51,52,53,54]. 

For a more detailed comparison of the *Eranthis* karyotypes, we conducted a classical molecular cytogenetic study on the localization of conserved ribosomal gene sequences. We tested a new oligonucleotide probe to identify tandem repeats of 45S rDNA, which obviously can be used for further studies on other taxa. We reported the location of 45S and 5S rDNA on the chromosomes of *E. cilicica*, *E. hyemalis*, and *E. longistipitata*, which confirm our results on the species-specific chromosomal organization of their genomes. The number of ribosomal DNA clusters and their distribution along chromosomes varied in related species such as *E. cilicica* and *E. hyemalis*. The genomes of *E. cilicica* and *E. longistipitata* contain one and two pairs of 5S rDNA clusters, respectively; the repeats in *E. hyemalis* were dispersed throughout the genome and could be found in almost all chromosomes of the set, except for a satellite pair of chromosomes. This shows that the genomes have undergone relevant changes in their evolution.

## 4. Materials and Methods

### 4.1. Plant Samples

Plant materials (seeds and tubers) of *Eranthis bulgarica*, *E. cilicica*, *E. hyemalis*, and *E. longistipitata* were collected during field investigations in Bulgaria, Turkey, Italy, Germany, Hungary, Kazakhstan, Uzbekistan, and Tajikistan during 2017–2021. The list of the samples examined is presented in Table 1. Herbarium specimens were deposited in the TK and NS herbaria (herbarium acronyms according to Thiers 2023, continuously updated).

### 4.2. Karyotype Analysis

Chromosome counts were conducted for 18 populations, while the karyotype analysis was conducted for 12 populations of *E. bulgarica*, *E. cilicica*, *E. hyemalis*, and *E. longistipitata* (Table 1 and Table 2). Somatic chromosomes of *Eranthis* were prepared from root meristems. Tubers and seeds were germinated at ~5 °C for 1–3 months. Newly formed 1–2 cm long roots were excised and pretreated in 0.5% colchicine solution for 3–4 h. Roots were fixed in a mixture of 96% ethanol and glacial acetic acid (3:1). Chromosomes stained with 1% aceto-hematoxylin were used for morphometric analyses. Staining and karyotyping were performed according to standard protocols [29,55]. Mitotic metaphase chromosome plates were studied using an Axio Star microscope (Carl Zeiss, Munich, Germany) and photographed using an Axio Imager A.1 microscope (Carl Zeiss, Munich, Germany) equipped with AxioCam MRc5 CCD-camera (Carl Zeiss, Munich, Germany) at 1000× magnification using AxioVision 4.7 software (Carl Zeiss, Munich, Germany) in the Laboratory for Ecology, Genetics and Environmental Protection (National Research Tomsk State University, Tomsk, Russia). KaryoType 2.0 software [56] was used for karyotyping, and Adobe Photoshop CS5 (Adobe Systems, San Jose, CA, USA) and Inkscape 0.92 (Boston, MA, USA) were used for image editing.

The measurements were performed on 5–9 metaphase plates per population. The symbols used to describe the karyotypes corresponded to those coined by Levan et al. (1964): m = median centromeric chromosome with arm ratio (r) of 1.0–1.7 (metacentric chromosome); sm = submedian centromeric chromosome with arm ratio of 1.7–3.0 (submetacentric chromosome); st = subterminal centromeric chromosome with arm ratio of 3.0–7.0 (subtelocentric chromosome); t = terminal centromeric chromosome with arm ratio of 7.0 and more (acrocentric chromosome); T = chromosome without obvious short arm (telocentric chromosome). Mean values of arm ratio (r), centromeric indices (CI), and relative chromosome length (RL) for each chromosome pair and total haploid length (THL) were determined. In addition, we calculated the coefficient of variation in chromosome length (CV_CL_) [47], coefficient of variation in centromeric index (CV_CI_) [47], and mean centromeric asymmetry (M_CA_) [46].

Given that the chromosome number (2*n*) and the basic chromosome number (*x*) were constant in the studied species, we slightly modified the approach proposed by Peruzzi and Altınordu [57] by directly considering only THL, CV_CL_, M_CA_, and CV_CI_ as variables in an exploratory principal component analysis (PCA). A linear discriminant analysis (LDA) was carried out to test the karyomorphometric diagnosability of the four species, considering them as a priori groups. Each variable was also subjected to univariate analysis (ANOVA followed by Tukey HSD test for multiple comparisons). All the analyses were carried out with the software PAST 4.14. [58,59], freely available online.

### 4.3. Oligonucleotide Probes

For the 5S rDNA localization we used the oligonucleotide DNA probe Oligo-5S rDNA (5′-TCA GAA CTC CGA AGT TAA GCG TGC TTG GGC GAG AGT AC-3′) based on the sequence of pTa794 probe of wheat (*Triticum aestivum* L.) [60]. The oligo-probe was labeled with the fluorescein amidite (FAM) from the 5′ end. For the 45S rDNA localization in Ranunculaceae, we developed and tested a new oligo-probe. To obtain a universal DNA sequence appropriate for the development of an oligonucleotide DNA probe, rDNA sequences of several plants were analyzed. Sequences of *Adonis amurensis* Regel & Radde (LC540384.1), *A. shikokuensis* Nishikawa & Koji Ito (LC540399.1), *Anemone cernua* var. *koreana* Y. Yabe ex Nakai (GU732647.1), *Coptis chinensis* Franch. (KC815158.1), *Pulsatilla turczaninovii* Krylov & Serg. (GU732649.1), and *Trollius chinensis* Bunge (AH006943.2), were chosen and downloaded from GenBank. Sequence alignment was performed using UGENE v38.1 (Unipro, Novosibirsk, Russia) with the ClustalW tool. A 121-nucleotide long consensus sequence corresponding to the 5.8S rRNA gene was identified, and its universality among flowering plants was confirmed with the NCBI BLAST analysis (https://blast.ncbi.nlm.nih.gov, accessed on 20 April 2021). The uniqueness of the identified sequence was verified by BLAST alignment against the *Arabidopsis thaliana* (L.) Heynh. genome. The analysis showed homology with the beta-glucosidase gene for the fragment from 63 to 82 nucleotides; therefore, a specific fragment from 13 to 62 nucleotides with a total length of 50 nucleotides was chosen for the DNA probe (designated as Oligo-5.8S rDNA-Ran). Probe nucleotide sequence: 5′-ACGGATATCTCGGCTCTTGCATCGATGAAGAACGTAGCGAAATGCGATAC-3′. This probe was labeled with tetramethylrhodamine (TAMRA) from the 5′ end. The oligonucleotide probes used in the study were synthesized in the “Evrogen” (Russia, Moscow).

### 4.4. Chromosome Preparation and Fluorescence In Situ Hybridization (FISH)

Newly formed 1–2 cm long roots were excised and pretreated in 0.5% colchicine solution for 3–4 h. Roots were fixed in a mixture of 96% ethanol and glacial acetic acid (3:1). Before squashing, the roots were transferred into 1% acetocarmine in 45% acetic acid for 15–30 min. The cover slips were removed after freezing in liquid nitrogen. The slides were dehydrated in 96% ethanol and then air dried.

Fluorescence in situ hybridization with oligo-probes was carried out according to the protocol published by Badaeva et al. [61]. Prior to hybridization, slides were treated with RNAse A (Sigma, Livonia, MI, USA) solution in moist chamber at 37 °C for 1 h. Chromosomes were stained with 2 ng/μL DAPI (4′,6-diamidino-2-2phenylindole) in antifade medium Vectashield (Vector Laboratories, Burlingame, CA, USA).

Chromosomal preparations were examined and the selected mitotic metaphase plates were photographed using an AxioImager Z.1 (Carl Zeiss, Germany) epifluorescent microscope with AxioVision 4.8 software (Carl Zeiss, Germany) and AxioCam MRm CCD-camera (Carl Zeiss, Germany) at 1000× magnification in the Laboratory for Ecology, Genetics and Environmental Protection (National Research Tomsk State University, Tomsk, Russia). Adobe Photoshop CS5 (Adobe Systems, San Jose, CA, USA) and Inkscape 0.92 (Boston, MA, USA) software were used for image editing.

## 5. Conclusions

Comparative karyotype analysis of four species of yellow-flowered *Eranthis* sect. *Eranthis*, i.e., *E. bulgarica*, *E. cilicica*, *E. hyemalis*, and *E. longistipitata* from different areas, was carried out. Despite the fact that some authors consider that *E. bulgarica* is a synonym of *E. hyemalis*, cytogenetic analysis showed clear differences between these taxa suggesting that they are two distinct species. Not all the species in the type section were included in the analysis. For further studies, it is planned to investigate the karyotypes of *E. iranica*, *E. kurdica*, and *E.* × *tubergenii* and compare them with *E. bulgarica*, *E. cilicica*, *E. hyemalis*, and *E. longistipitata*. In addition, the taxonomy and phylogenetic relationships of the recently described species (*E. bulgarica*, *E. iranica*, and *E. kurdica*) need to be verified by integrating morphological, molecular phylogenetic, and cytogenetic approaches.

## Figures and Tables

**Figure 1 plants-13-00047-f001:**
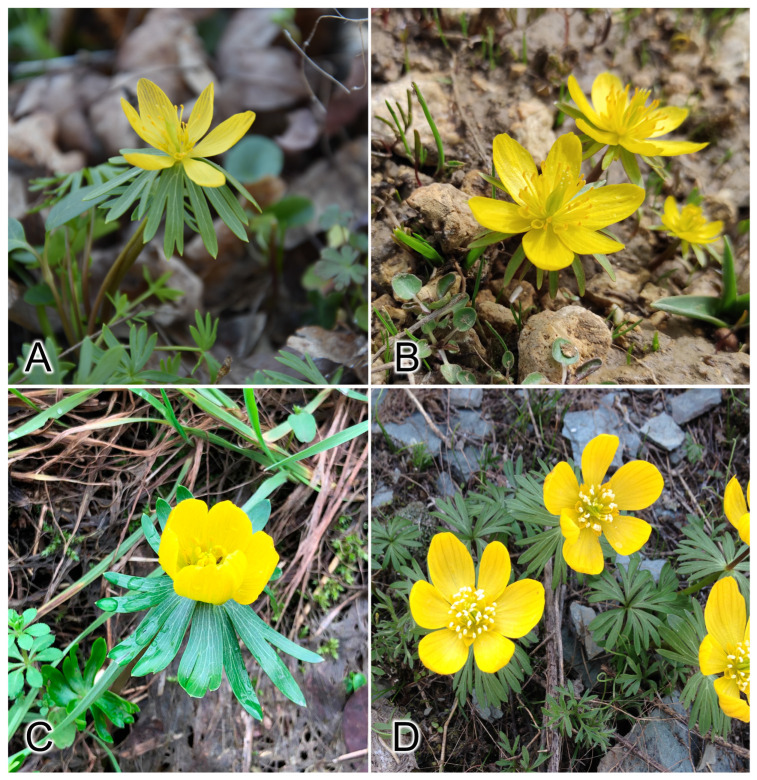
The studied species of yellow-flowered *Eranthis* sect. *Eranthis* (**A**) *E. bulgarica*, (**B**) *E. cilicica*, (**C**) *E. hyemalis*, (**D**) *E. longistipitata*.

**Figure 2 plants-13-00047-f002:**
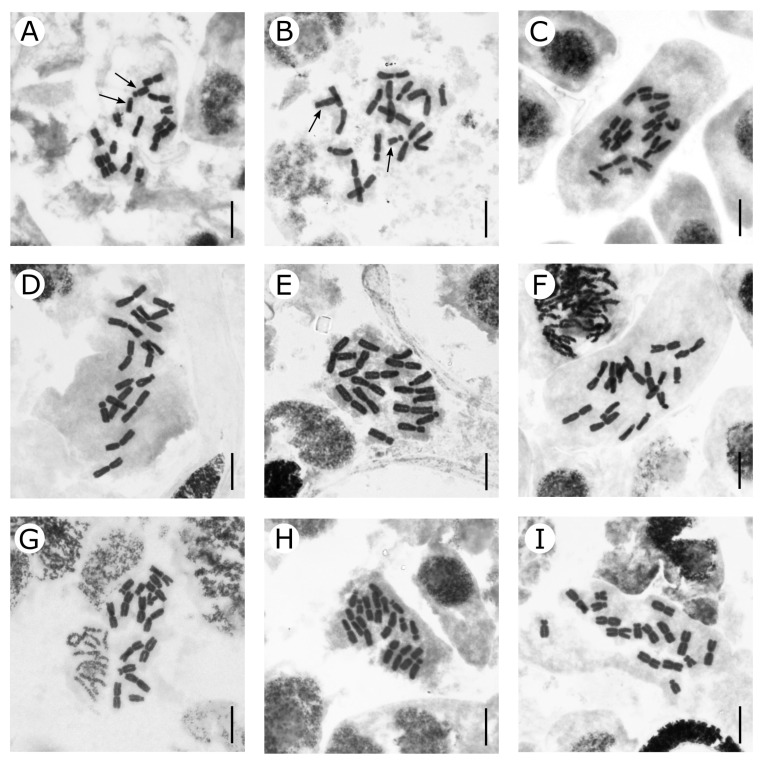
Mitotic metaphase chromosome plates of yellow-flowered *Eranthis* sect. *Eranthis.* (**A**)—*E. bulgarica* (pop. 1), 2*n* = 16; (**B**)*—E. bulgarica* (pop. 2), 2*n* = 16; (**C**)—*E. cilicica* (pop. 3), 2*n* = 16; (**D**)—*E. hyemalis* (pop. 9), 2*n* = 16; (**E**)—*E. hyemalis* (pop. 11), 2*n* = 16; (**F**)—*E. hyemalis* (pop. 12), 2*n* = 16; (**G**)—*E. longistipitata* (pop. 14), 2*n* = 16; (**H**)—*E. longistipitata* (pop. 15), 2*n* = 16; (**I**)—*E. longistipitata* (pop. 17), 2*n* = 16. Arrows point at heteromorphic chromosome pair. Scale bars = 10 µm.

**Figure 3 plants-13-00047-f003:**
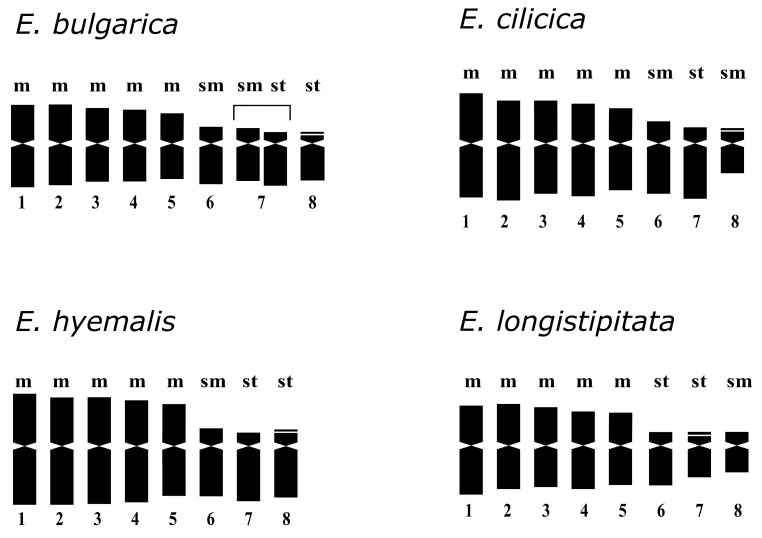
Haploid idiograms of yellow-flowered *Eranthis* sect. *Eranthis* species. m—metacentric chromosome; sm—submetacentric chromosome; st—subtelocentric chromosome; 1–8—chromosome pairs.

**Figure 4 plants-13-00047-f004:**
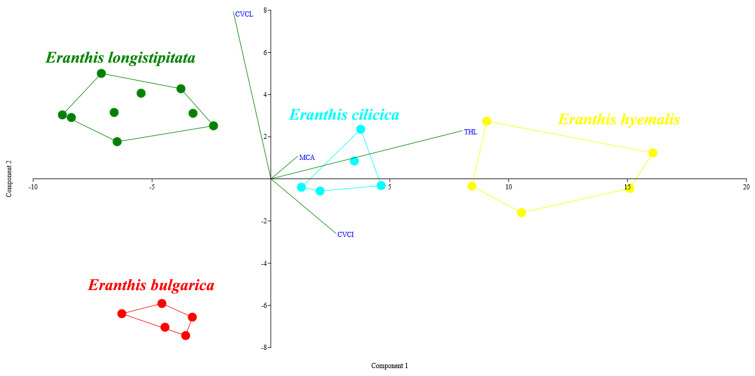
PCA biplot based on the four karyological parameters (THL, CV_CL_, M_CA_, CV_CI_) of yellow-flowered *Eranthis* sect. *Eranthis* species. The accessions (metaphasic plates) of each species are represented with different colors. Green lines length is proportional to the relative contribution of each of the four karyological parameters in displacing the accessions in the bidimensional space.

**Figure 5 plants-13-00047-f005:**
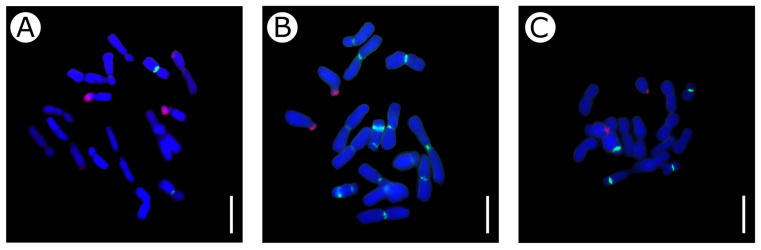
The 45S (red, TAMRA) and 5S (green, FAM) ribosomal DNA distribution on metaphase chromosomes of *Eranthis cilicica*, pop. 3, 2*n* = 16 (**A**), *E. hyemalis*, pop. 10, 2*n* = 16 (**B**), and *E. longistipitata*, pop. 17, 2*n* = 16 (**C**). DAPI chromosome staining—blue. Scale bars = 10 µm.

**Table 1 plants-13-00047-t001:** Chromosome number in yellow-flowered *Eranthis* sect. *Eranthis* from different localities.

Pop.	Species	Voucher Information	CN, 2*n*
1	*E. bulgarica*	Bulgaria, Vidin District, Vrashka Chuka Peak, xerothermal belt of oak forests, 632 m, 43°50′14.2″ N 22°22′30.3″ E, 12.03.2019, A.N. Tashev, BU2019-2	16
2	*E. bulgarica*	Bulgaria, Vidin District, Vrashka Chuka Peak, xerothermal belt of oak forests, 678 m, 43°49′54.8″ N 22°17′0.2″ E, 16.03.2021, A.N. Tashev, S. Bancheva, BU2021-5.3	16
3	*E. cilicica*	Turkey, Kahramanmaraş Province, Göksun District, Delihöbek Dagi, mountain steppe, 2115 m, 37°53′ N 36°41′ E, 29.04.2019, T. Ertuğrul, TU2019-1	16
4	*E. cilicica*	Turkey, Konya Province, between Taşkent–Başyayla, after Feslikan Plateau, high mountain steppe with *Ornithoghalum lanceolatum*, *Anemone blanda*, *Tulipa armena*, *Ranunculus* sp., 1726 m, 36°50′45″ N 32°33′36″ E, 2021, A.S. Erst, T.V. Erst, O. Çeçen, Z. Aytac, TU2021-4	16
5	*E. cilicica*	Turkey, Karaman Province, between Taşkent–Başyayla, 10 km from Basuala, high mountain steppe with *Ornithoghalum lanceolatum*, *Tulipa cinnaborina*, *Acantholimon venestum*, *Corydalis erdelii*, *Fritillaria pinardii*, *Anemone* blanda, 1807 m, 36°47′20″ N 32°37′54″ E, 2021, A.S. Erst, T.V. Erst, O. Çeçen, Z. Aytac, TU2021-5	16
6	*E. cilicica*	Turkey, Mersin Province, Anamur District, Tamtır Plateau, high mountain stony steppe with *Corydalis* sp., *Anemone blanda*, *Fritillaria pinardi*, *Ornithogalum platyphyllum*, 1889 m, 36°19′36″ N 32°43′16″ E, 2021, A.S. Erst, T.V. Erst, O. Çeçen, Z. Aytac, TU2021-11	16
7	*E. hyemalis*	Italy, Impruneta, Villa Le Rose (Firenze), olive grove, 107 m, 43°43′15.5″ N 11°13′36.2″ E, 15.02.2020, A.S. Erst, T.V. Erst, L. Pinzani, IT2020-1	16
8	*E. hyemalis*	Italy, Pontassieve, Villa di Grignano (Firenze), olive grove along the street, 300 m, 43°48′23.7″ N 11°27′32.1″ E, 15.02.2020, A.S. Erst, T.V. Erst, L. Pinzani, IT2020-4	16
9	*E. hyemalis*	Italy, Via di Roncrio, Bologna, mixed deciduous wood near road, 123 m, 44°27′42.1″ N 11°20′12.9″ E, 16.02.2020, A.S. Erst, T.V. Erst, L. Pinzani, IT2020-5	16
10	*E. hyemalis*	Italy, Franzone La Ponte, Lombardia, Goldferenzo (Pavia), along the road, 238 m, 44°58′11.4″ N 9°17′31.1″ E, 17.02.2020, A.S. Erst, T.V. Erst, L. Pinzani, IT2020-8	16
11	*E. hyemalis*	Germany, Rheinland-Pfalz, Dörrmoschel, cemetery along the street, 362 m, 49°37′07.9″ N 7°45′07.1″ E, 12.02.2020, I. Vogler, C. Rosche, GE2020-3	16
12	*E. hyemalis*	Hungary, Nagykapornak, park with remain *Quercus–Carpinus* forest, 163 m, 46°49′13″ N 16°59′33″ E, 15.02.2020, A. Mesterházy, HU2020-1	16
13	*E. hyemalis*	Hungary, Aszófő, *Quercus–Carpinus* forest in the valley, 150 m, 46°56′18″ N 17°49′31″ E, 18.02.2020, A. Mesterházy, HU2020-2	16
14	*E. longistipitata*	Kazakhstan, western part of the Kirghizsky Ridge, Botamoynak Mountains, near Taraz City, 900 m, 42°54′26″ N 71°32′09″ E, 24.03.2017, V. Kolbinzev, KAZ2017-1	16
15	*E. longistipitata*	Tajikistan, Khatlon Oblast, Muminobod district, Hazrati-Shoh pass, Childukhtaron mountain, blackwood, 38°18′11.5″ N 70°09′57.7″ E, 10.04.2020, M.T. Boboev, S.B. Yoqubov, TJ2020-1	16
16	*E. longistipitata*	Uzbekistan, Andijan region, Khojaabad region, east-southeastern part of the Fergana Valley, Kyrtashtau mountains, near Imamat village, mossy stony slope, 910 m, 40°32′27″ N, 72°36′28″ E, 12.03.2020, T.Kh. Makhkamov, D.A. Krivenko, O.T. Turginov, O.A. Chernysheva, UZB2020-1	16
17	*E. longistipitata*	Uzbekistan, Tashkent region, Bostanlyk district, Western Tan-Shan, north-western part of the Chatkal ridge, foot of the Big Chimgan mountain, between Galvasay and Mramornaya rivers, on the road from Uchterek tract to Chimgan tract, bushy slope, 1690 m, 41°31′05″ N 69°59′15″ E, 16.03.2020, D.A. Krivenko, O.A. Chernysheva, T.Kh. Makhkamov, UZB2020-9	16
18	*E. longistipitata*	Uzbekistan, Jizzakh region, Zaamin district, Western Pamiro-Alai, Gissar-Alai, northern macroslope of the Turkestan ridge, Zaamin forestry enterprise, Usman tract, mountain slope, 1450 m, 39°43′26.1″ N 68°27′54.0″ E, 20.03.2020, T.Kh. Makhkamov, UZB2020-10	16

Notes: Pop.—population number; CN—chromosome number.

**Table 2 plants-13-00047-t002:** Karyotype formulas in yellow-flowered *Eranthis* sect. *Eranthis*.

Species	Population Number/Voucher	Karyotype Formula
*E. bulgarica*	Pop. 1, BU2019-2	10m + 3sm + 1st + 2st^sat^
	Pop. 2, BU2021-5.3	10m + 3sm + 1st + 2st^sat^
*E. cilicica*	Pop. 3, TU2019-1	10m + 4sm + 2st^sat^
	Pop. 4, TU2021-4	10m + 4sm + 2st^sat^
*E. hyemalis*	Pop. 7, IT2020-1	10m + 2sm + 2st + 2st^sat^
	Pop. 8, IT2020-4	10m + 2sm + 2st + 2st^sat^
	Pop. 9, IT2020-5	10m + 2sm + 2st + 2st^sat^
	Pop. 10, IT2020-8	10m + 2sm + 2st + 2st^sat^
	Pop. 11, GE2020-3	10m + 2sm + 2st + 2st^sat^
	Pop. 12, HU2020-1	10m + 2sm + 2st + 2st^sat^
*E. longistipitata*	Pop. 14, KAZ2017-1	10m + 2sm + 2st + 2st^sat^
	Pop. 15, TAJ 2020-1	10m + 2sm + 2st + 2st^sat^
	Pop. 17, UZB2020-9	10m + 2sm + 2st + 2st^sat^

Notes: m—metacentric chromosome; sm—submetacentric chromosome; st—subtelocentric chromosome; ^sat^—satellite chromosome.

**Table 3 plants-13-00047-t003:** Karyomorphological parameters in yellow-flowered *Eranthis* sect. *Eranthis*.

Species	Pair	CL, µm	r	CI, %	RL, %	Type	THL	M_CA_	CV_CL_	CV_CI_
*E. bulgarica* (Pop. 1)	1	6.99 ± 0.23	1.14 ± 0.06	46.67	7.75	m	45.12 ± 0.96	21.13 ± 0.67	18.57 ± 0.63	27.76 ± 0.83
	2	6.71 ± 0.29	1.07 ± 0.05	48.49	7.44	m				
	3	6.27 ± 0.17	1.07 ± 0.04	48.26	6.95	m				
	4	6.14 ± 0.21	1.12 ± 0.06	47.29	6.80	m				
	5	5.59 ± 0.27	1.13 ± 0.05	46.93	6.20	m				
	6	4.88 ± 0.23	2.48 ± 0.13	28.76	5.41	sm				
	7	4.42 ± 0.04 4.59 ± 0.11	2.40 ± 0.18 3.58 ± 0.31	29.44 21.97	4.90 5.10	sm st				
	8	4.02 ± 0.15	3.28 ± 0.13	23.42	4.45	st^sat^				
*E. cilicica* (Pop. 3)	1	8.35 ± 0.42	1.06 ± 0.03	48.48	7.60	m	54.90 ± 1.91	22.86 ± 0.98	22.11 ± 1.05	24.19 ± 1.69
	2	8.26 ± 0.28	1.23 ± 0.07	44.96	7.52	m				
	3	7.86 ± 0.23	1.13 ± 0.06	47.09	7.16	m				
	4	7.69 ± 0.26	1.36 ± 0.09	42.53	7.00	m				
	5	6.97 ± 0.45	1.36 ± 0.07	42.42	6.35	m				
	6	6.02 ± 0.36	2.39 ± 0.32	29.82	5.49	sm				
	7	6.02 ± 0.38	3.51 ± 0.43	22.37	5.48	st^sat^				
	8	3.73 ± 0.16	2.25 ± 0.24	30.97	3.40	sm				
*E. hyemalis* (Pop. 9)	1	9.75 ± 0.44	1.07 ± 0.03	48.34	7.90	m	61.71 ± 2.72	24.10 ± 1.55	22.40 ± 1.64	31.87 ± 2.31
	2	9.24 ± 0.47	1.19 ± 0.08	45.82	7.48	m				
	3	9.13 ± 0.40	1.08 ± 0.05	48.08	7.39	m				
	4	8.49 ± 0.39	1.24 ± 0.10	44.80	6.88	m				
	5	7.98 ± 0.58	1.13 ± 0.09	46.97	6.46	m				
	6	5.97 ± 0.33	2.48 ± 0.16	28.72	4.84	sm				
	7	5.59 ± 0.28	4.58 ± 0.83	18.28	4.53	st^sat^				
	8	5.57 ± 0.44	3.59 ± 0.45	21.83	4.51	st				
*E. longistipitata* (Pop. 15)	1	7.68 ± 0.54	1.23 ± 0.07	44.96	8.27	m	46.44 ± 2.24	22.31 ± 1.14	27.93 ± 1.21	24.45 ± 0.93
	2	7.35 ± 0.30	1.08 ± 0.04	48.16	7.91	m				
	3	6.95 ± 0.31	1.16 ± 0.07	46.28	7.48	m				
	4	6.78 ± 0.41	1.30 ± 0.08	43.58	7.30	m				
	5	6.14 ± 0.43	1.18 ± 0.08	46.04	6.61	m				
	6	4.45 ± 0.36	3.16 ± 0.19	24.41	4.79	st				
	7	3.82 ± 0.22	3.14 ± 0.14	24.40	4.12	st^sat^				
	8	3.27 ± 0.31	2.05 ± 0.19	33.00	3.52	sm				

Notes: Pair—chromosome pair; CL—chromosome length (M ± SD); r—arm ratio (M ± SD); CI—centromeric index; RL—relative chromosome length; m—metacentric chromosome; sm—submetacentric chromosome; st—subtelocentric chromosome; ^sat^—satellite chromosome; Type—morphological chromosome type; THL—total length of haploid chromosome set; M_CA_—mean centromeric asymmetry (M ± SD); CV_CL_—coefficient of variation in chromosome length (M ± SD); CV_CI_—coefficient of variation in centromeric index (M ± SD).

## Data Availability

All data generated or analysed during this study are included in this article.

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
