# Peer review of "Karyotypes and Physical Mapping of Ribosomal DNA with Oligo-Probes in Eranthis sect. Eranthis (Ranunculaceae)"

_plants, 2023, doi:10.3390/plants13010047_

Round 1
Reviewer 1 Report
Comments and Suggestions for Authors
This paper had done comparative karyotype analysis of four out of the six species of yellow-flowered Eranthis sect. Eranthis from different areas. The study for the chromosome number of E. bulgarica, E. cilicica, and E. longistipitata has been conducted for the first time and produce a significant results. In general, this manuscript is well written and some results are novel and have significant theoretical application values. I have only one minor comments:
Minor comment:
1. Reference need to be updated. There is only one reference cited from 2023.
Author Response
Thank you very much for taking the time to review this manuscript. Please find the detailed responses below and the corresponding revisions/corrections highlighted/in track changes in the re-submitted files.

Reviewer 2 Report
Comments and Suggestions for Authors
The article "Karyotypes and Physical Mapping..." by Mitrenina et al. uses karyotyping and FISH mapping of 5S and 5.8S oligonucleotide probes to present information on Eranthis sect. Eranthis chromosomes from collected populations. The figures, data within Tables and the conclusions drawn are both detailed and acceptable as presented in the article. The Introduction is a bit confusing and harder to read than the other sections of the paper. Some suggestions/edits have been placed in the edited attached file.

The English grammar on the manuscript is fine. Organization, a bit more explanation in the Introduction would make the manuscript clearer.
Author Response

(The authors gave the same response as above.)

Reviewer 3 Report
Comments and Suggestions for Authors Authors have published studies regarding this topic, so the original part only corresponds to the description of comparative Karyotypes of four out of the six species of yellow-flowered Eranthis sect genus; being relevant to those who study the genre Eranthis sect.; by example: Karyotype and genome size variation in white-flowered Eranthis sect. Shibateranthis (Ranunculaceae). PhytoKeys 187: 207-22710.3897/phytokeys.187.75715
The study contributes to the thematic area with data from chromosome analysis of five species of the same genus (Eranthis sect.) in comparison with the publication of comparative karyotype analysis and genome size of six out of eight white-flowered species of same genus (Eranthis sect.).
The tables and figures are well represented, except figure 5 which is poorly readable and dark.
The main question or objective of the study that the research addresses is not well defined, the authors must define the reason for the study at the end of the Introduction
The authors should consider in the methodology the use of flow cytometry with propidium iodide (PI) staining to determine the absolute DNA content. What further controls should be considered? Additional controls such as determining the genome size of the six white-flowered Eranthis species should be considered.
Indicate at the end of the "Introduction" the objective of the study, not the results obtained (an oligoprobe corresponding to a fragment of the 5.8S rRNA gene was developed and successfully tested).
The relevance of the study must be firmly established, because it develops the chromosomal analysis of the six species of the Sect. Eranthis, what is the importance of carrying out this study.
Author Response

(The authors gave the same response as above.)
